# Study protocol for the Innovative Support for Patients with SARS-COV-2 Infections Registry (INSPIRE): A longitudinal study of the medium and long-term sequelae of SARS-CoV-2 infection

Kelli N. O'Laughlin[ID][1,2]*, Matthew Thompson[3], Bala Hota[4], Michael Gottlieb[5], Ian D. Plumb[6], Anna Marie Chang[7], Lauren E. Wisk[8], Aron J. Hall[6], Ralph C. Wang[9], Erica S. Spatz[10], Kari A. Stephens[3], Ryan M. Huebinger[5], Samuel A. McDonald[11], Arjun Venkatesh[12], Nikki Gentile[3], Benjamin H. Slovis[7], Mandy Hill[ID][13], Sharon Saydah[6], Ahamed H. Idris[11], Robert Rodriguez[9], Harlan M. Krumholz[ID][10], Joann G. Elmore[8], Robert A. Weinstein[ID][4,14], Graham Nichol[1,15], for the INSPIRE Investigators[¶]

1 Department of Emergency Medicine, University of Washington, Seattle, WA, United States of America, 2 Department of Global Health, University of Washington, Seattle, WA, United States of America, 3 Department of Family Medicine, University of Washington, Seattle, WA, United States of America, 4 Division of Infectious Diseases, Department of Internal Medicine, Rush University Medical Center, Chicago, IL, United States of America, 5 Department of Emergency Medicine, Rush University Medical Center, Chicago, IL, United States of America, 6 Division of Viral Diseases, Centers for Disease Control and Prevention, Respiratory Viruses Branch, Atlanta, GA, United States of America, 7 Department of Emergency Medicine, Thomas Jefferson University, Philadelphia PA, United States of America, 8 Department of Medicine, David Geffen School of Medicine at UCLA, Los Angeles, CA, United States of America, 9 Department of Emergency Medicine, University of California, San Francisco, CA, United States of America, 10 Section of Cardiovascular Medicine, Yale School of Medicine, New Haven, CT, United States of America, 11 Department of Emergency Medicine and Clinical Informatics Center, UT Southwestern, Dallas, TX, United States of America, 12 Department of Emergency Medicine, Yale School of Medicine, New Haven, CT, United States of America, 13 Department of Emergency Medicine, UTHealth McGovern Medical School, Houston, TX, United States of America, 14 Department of Medicine, Cook County Health, Chicago, IL, United States of America, 15 Departments of Medicine, University of Washington, Seattle, WA, United States of America

¶ Membership of the INSPIRE Investigators is listed in the S1 Appendix.
* kolaugh@uw.edu

**Funding:** The Innovative Support for Patients with SARS-COV-2 Infections (INSPIRE) Registry is

## Abstract

### Background

Reports on medium and long-term sequelae of SARS-CoV-2 infections largely lack quantification of incidence and relative risk. We describe the rationale and methods of the Innovative Support for Patients with SARS-CoV-2 Registry (INSPIRE) that combines patient-reported outcomes with data from digital health records to understand predictors and impacts of SARS-CoV-2 infection.

### Methods

INSPIRE is a prospective, multicenter, longitudinal study of individuals with symptoms of SARS-CoV-2 infection in eight regions across the US. Adults are eligible for enrollment if they are fluent in English or Spanish, reported symptoms suggestive of acute SARS-CoV-2

funded by the Centers for Disease Control and Prevention (CDC, www.cdc.gov), National Center of Immunization and Respiratory Diseases (NCIRD) (contract number: 75D30120C08008; co-PIs Bala Hota, MD, and Robert A. Weinstein, MD). Partners from the CDC (IDP, AJH, SS) assisted with study design, the preparation of this protocol manuscript and the decision to publish this manuscript. The findings and conclusions in this report are those of the authors and do not necessarily represent the official position of the Centers for Disease Control and Prevention (CDC). No others sponsors or funders (other than the named authors) played any role in the study design, data collection and analysis, decision to publish, or preparation of the manuscript. Additionally, I read the journal's policy and the authors of this manuscript have the following financial disclosures: JGE is an Editor for UpToDate on topics related to SARS-CoV-2 (https://www.uptodate.com). GN receives salary support from Leonard A Cobb~ Medic One Foundation (https://www.mediconefoundation.org/about/who-we-are/) via the University of Washington. He is a consultant to ZOLL Circulation Inc. (https://www.zoll.com/contact/careers-at-zoll/tms), ZOLL Medical Corporation (https://www.zoll.com/contact/careers-at-zoll/resuscitation), Cellphire Therapeutics Inc. (https://www.cellphire.com/) Kestra Medical Technologies (https://kestramedical.com/), Abiomed Inc. (https://www.abiomed.com/), CPR Therapeutics Inc. (https://www.cprtherapeutics.com/). HMK received expenses and/or personal fees from UnitedHealth (https://www.uhc.com/), Element Science (https://www.elementscience.com/), Aetna (https://www.aetna.com/), Facebook (https://www.facebook.com/), the Siegfried and Jensen Law Firm (https://www.siegfriedandjensen.com/), Arnold and Porter Law Firm (https://www.theworldlawgroup.com/member-firms/arnold-porter), Martin/Baughman Law Firm (https://www.martinbaughman.com/practice-areas/), and F-Prime (https://fprimecapital.com/). He is a co-founder of Refactor Health (https://www.refactorhealth.com/) and had grants and/or contracts from the Centers for Medicare & Medicaid Services (https://www.cms.gov/), the U. S. Food and Drug Administration (https://www.fda.gov/), and Johnson & Johnson (https://www.jnj.com/). MT provided consulting as a paid medical advisor to Visby Medical (https://www.visbymedical.com/) and Roche Molecular Diagnostics (https://diagnostics.roche.com/) which both produce laboratory tests for COVID-19. The authors not otherwise listed here have no financial disclosures.

**Competing interests:** I read the journal's policy and the authors of this manuscript have the

infection, and if they are within 42 days of having a SARS-CoV-2 viral test (i.e., nucleic acid amplification test or antigen test), regardless of test results. Recruitment occurs in-person, by phone or email, and through online advertisement. A secure online platform is used to facilitate the collation of consent-related materials, digital health records, and responses to self-administered surveys. Participants are followed for up to 18 months, with patient-reported outcomes collected every three months via survey and linked to concurrent digital health data; follow-up includes no in-person involvement. Our planned enrollment is 4,800 participants, including 2,400 SARS-CoV-2 positive and 2,400 SARS-CoV-2 negative participants (as a concurrent comparison group). These data will allow assessment of longitudinal outcomes from SARS-CoV-2 infection and comparison of the relative risk of outcomes in individuals with and without infection. Patient-reported outcomes include self-reported health function and status, as well as clinical outcomes including health system encounters and new diagnoses.

## Results

Participating sites obtained institutional review board approval. Enrollment and follow-up are ongoing.

## Conclusions

This study will characterize medium and long-term sequelae of SARS-CoV-2 infection among a diverse population, predictors of sequelae, and their relative risk compared to persons with similar symptomatology but without SARS-CoV-2 infection. These data may inform clinical interventions for individuals with sequelae of SARS-CoV-2 infection.

## Introduction

The COVID-19 pandemic is associated with considerable morbidity and mortality. As of December 2021, > 51 million COVID-19 cases and > 805,000 attributed deaths have been detected in the USA [1]. Globally, > 276 million COVID-19 cases and > 5.3 million attributed deaths have been reported [2]. The clinical course of acute COVID-19 is well-described [3–7]. According to the Centers for Disease Control, post-acute COVID-19 is defined as emergent, recurring or persistent symptoms occurring ≥ 4 weeks following acute infection with COVID-19 (https://www.cdc.gov/coronavirus/2019-ncov/long-term-effects/index.html). Other sources describe post-acute COVID-19 as persistence of symptoms or development of sequelae after 3 or 4 weeks from the onset of acute symptoms of COVID-19 [8–10]. Some have divided the post-acute time period into subacute period (4–12 weeks) beyond acute COVID-19, and a chronic or post-COVID-19 syndrome (> 12 weeks), which includes symptoms persisting or present not attributable to alternative diagnoses [8, 11]. Information on post-acute COVID-19 and long-term sequelae of SARS-CoV-2 infection has continued to emerge [12–15].

To characterize post-COVID-19 syndromes better, there is an urgent need greater diversity in the study population to allow for representativeness and generalizability, to set objective outcomes that are not limited to symptoms but include illness events and clinical events, and to include SARS-CoV-2 negative individuals to ensure other effects of the pandemic are considered in the analysis (e.g. impact on livelihoods, mental health, food security, and mobility)

following competing interests: HMK is co-founder for Hugo Health; he is not employed by Hugo Health and he receives no salary support from Hugo Health. Hugo Health is a vendor to support INSPIRE study operations and is not the funder of this study or the subject of the investigation. This does not alter our adherence to PLOS ONE policies on sharing data and materials.

[16]. Such work will help clinicians know what post-infectious sequelae to expect and who is at increased risk, will help ensure research findings can be compared across studies, and will move us towards addressing critical health care response needs.

To accelerate research during this critical time in the global COVID-19 pandemic, our research consortium designed a prospective longitudinal study to use patient-reported information linked to real-world data through the Innovative Support for Patients With SARS-COV2 Infections Registry (INSPIRE) hosted on a secure online platform (Hugo, Hugo Health LLC, Guilford, CT) which imports health information from various sources with permission from participants. In this study, researchers follow a sample of individuals under investigation for SARS-CoV-2 over time, to collect patient-reported outcomes, interactions with the medical system (i.e., clinic visits, hospitalizations, laboratory test and medications prescribed) and outcomes of care as reported in the electronic medical record (EMR). The goals of this research are to understand the medium- and long-term sequelae of symptomatic SARS-CoV-2 infection, to describe predictors of sequelae as reported by individuals and as recorded in their EMR, and to assess the intersection of long-term sequelae of COVID-19 with other previously defined syndromes with overlapping features such as myalgic encephalomyelitis/chronic fatigue syndrome (ME/CFS).

## Methods

### Study design

This is a prospective, multicenter, longitudinal cohort study of individuals with acute symptoms consistent with SARS-CoV-2, including those with positive and negative diagnostic SARS-CoV-2 tests to compare those with and without SARS-CoV-2 infection (ClinicalTrials. gov Identifier: NCT04610515) [17].

### Setting

Participants are enrolled from one of eight regions across the United States led by investigators at Rush University (Chicago, Illinois), Yale University (New Haven, Connecticut), the University of Washington (Seattle, Washington), Thomas Jefferson University (Philadelphia, Pennsylvania), the University of Texas Southwestern (Dallas, Texas), the University of Texas, Houston (Houston, Texas), the University of California, San Francisco (San Francisco, California) and the University of California, Los Angeles (Los Angeles County, California). The recruitment areas vary in terms of the population served to allow for ethnic and geographic diversity among study participants. As enrollment is through the internet and can be completed without assistance of research staff support, participants include those who have never been to the clinic or hospital for their symptoms and others who have been to the emergency department or hospitalized. Additionally, participants with any digital health portal can enroll in this study; participants' health portals need not be those directly linked to the academic institutions listed above.

### Participants

This study includes adult patients who are under clinical investigation for possible SARS-CoV-2 infection and who meet inclusion criteria outlined in Table 1. We include individuals with symptomatic presentation to determine comparative frequency of outcomes amongst symptomatic individuals with and without SARS-CoV-2 infection: symptomatic individuals presenting with covid-like illness who test negative will act as controls for symptomatic individuals who have positive tests for SARS-CoV-2. Asymptomatic individuals are not eligible,

**Table 1. Inclusion and exclusion criteria.**

| Inclusion criteria: |
| --- |
| a) Fluent in English or Spanish |
| b) Age 18 years or older |
| c) At least one self-reported symptom(s) suggestive of acute SARS-CoV-2 infection [19] |
| d) Tested for SARS-CoV-2 with any FDA-approved or authorized viral test (i.e., nucleic acid amplification test or antigen test [20]) within 42 days of enrollment |
| Exclusion criteria: |
| a) Unable to provide informed consent |
| b) Study team unable to confirm the result of a diagnostic test for SARS-CoV-2 |
| c) Does not have access to a hand-held device or computer that would allow for digital participation in the study |
| d) Lawfully imprisoned while participating in the study |

given the anticipated lower rates of outcomes among asymptomatic individuals who test positive for SARS-CoV-2, and the significant heterogeneity in reasons for testing in asymptomatic individuals, such as screening for social, educational, occupational reasons and prior to routine clinical procedures. Individuals who self-report symptoms suggestive of acute SARS-CoV-2 infection, and are tested for SARS-CoV-2 within the last 42 days, are eligible to participate. From study initiation through the summer of 2021, symptoms of acute SARS-CoV-2 were defined using the COVID-19 clinical criteria case definition [18]; beginning August of 2021, in order to capture less symptomatic individuals, the inclusion criteria were revised to require only one symptom among those listed in Table 2. Individuals for whom a SARS-CoV-2 test result cannot be confirmed are not eligible to participate in the study. Efforts are made to recruit participants from across the spectrum of COVID-19 illness severity, including individuals from outpatient (e.g., drive through testing with self-reported symptoms) to inpatient settings, and those cared for in intensive care units. We initially used a 3:1 case/control enrollment ratio to oversample those who are positive for SARS-CoV-2 on testing, while still

**Table 2. List of COVID-19 like symptoms used to determine study enrollment eligibility.**

| *Participants are asked, "Since you first felt sick with COVID-19 like symptoms, have you had any of the following?"* |
| --- |
| Note: presence of at a least one symptom is required for study participation |
| • Fever (>100.4F [38C])<br>• Feeling hot or feverish<br>• Chills<br>• Repeated shaking with chills<br>• More tired than usual<br>• Muscle aches<br>• Joint pains<br>• Runny nose<br>• Sore throat<br>• A new cough, or worsening of a chronic chough<br>• Shortness of breath<br>• Wheezing<br>• Pain or tightness in your chest<br>• Palpitations<br>• Nausea or vomiting<br>• Headache<br>• Hair loss<br>• Abdominal pain<br>• Diarrhea (>3 loose/looser than normal stools/24 hours)<br>• Decreased smell or change in smell<br>• Decreased taste or change in taste |

ensuring an adequate control cohort for comparison. However, given the heterogeneity in baseline characteristics among study subjects, we modified the enrollment to support a 1:1 case/control ratio, which will better enable the comparison of 'like' cases with 'like' controls.

**Participant identification and enrollment.** Methods used to recruit potentially eligible participants vary by site, although each site applies the same eligibility criteria described in Table 1. Most sites screen for eligible participants among those tested for SARS-CoV-2 infection. We seek to enroll participants as close to their initial date of SARS-CoV-2 testing as possible in order to reduce recall bias. Participant identification and recruitment methods include: i) participants learn of the study from a poster, brochure, or social media advertisement and/ or ii) research staff identify potentially eligible individuals and reach out to them in-person, over the phone (e.g., text or call), or by e-mail to invite them to enroll. In some instances, members of the study team access the EMR to screen for eligible individuals based on SARS-CoV-2 testing and reason for testing. In other cases, contact information are obtained from organizations conducting SARS-CoV-2 testing. The recruitment methods used at each site are based on local IRB approval and practical considerations. Regardless of how the individual is recruited, all participants must enroll through the online portal. Individuals may access this portal through any device that connects to the internet (e.g., smart phone, tablet, computer).

Though minor differences exist across sites, the study eligibility criteria, online enrollment, and data collection methods are identical, which allows for compilation and comparison of data across sites.

## Variables and outcomes

Patient-reported outcomes include self-reported disease-specific and health status outcomes (Table 3 and S1 Appendix) [21–23]. Health care utilization and clinical events are extracted from the EMR data via Hugo, ensuring uniform variable definitions across participating sites.

**Table 3. Summary of survey variables collected, instrument sources, and schedule of survey delivery.**

| Variable Type | Instrument Source | Survey schedule | | | | | | | |
|---|---|---|---|---|---|---|---|---|---|
| | | 0 (Pre-Enrollment) | 0 (Baseline) | 3 | 6 | 9 | 12 | 15 | 18 |
| **Screening questions** | Designed for INSPIRE | X | | | | | | | |
| **Socio-demographics** | CDC Patient Under Investigation | X | | | | | | | |
| **Testing information** | CDC Patient Under Investigation | | X | | | | | | |
| **Visits to healthcare facilities** | Designed for INSPIRE | | X | X | X | X | X | X | X |
| **Symptom check** | CDC Patient Under Investigation, case studies | | X | X | X | X | X | X | X |
| **Social determinants of health** | CMS Accountable Health Communities Health-Related Social Needs Screening Tool | | X | | | | | | |
| **Physical/mental health** | PROMIS-29 v2.1 | | X | X | X | X | X | X | X |
| **Cognitive Function** | PROMIS Cognitive SF 8a | | X | X | X | X | X | X | X |
| **Return to work/activity** | Designed for INSPIRE | | | X | X | X | X | X | X |
| **Post-infectious seq** | Modified Medical Research Council Dyspnea Scale, Cough Evaluation Test | | X | X | X | X | X | X | X |
| **PTSD** | Primary Care-PTSD-5 | | X | X | X | X | X | X | X |
| **Exercise** | Exercise vital sign | | X | X | X | X | X | X | X |
| **Fatigue symptoms** | CDC Short Symptom Screener | | X | X | X | X | X | X | X |
| **Severity of Illness** | Designed for INSPIRE | | | X | X | X | X | X | X |
| **Vaccine information** | Designed for INSPIRE | | X | X | X | X | X | X | X |
| **Comorbidities** | Designed for INSPIRE | | X | | | | | | |

SARS-CoV-2 Symptom Questionnaire—derived from the Centers for Disease Control and Prevention (CDC) Person Under Investigation for SARS-CoV-2 survey (Table 2) [24]. This is not scored and has no predefined cutoff for the likelihood of COVID-19 illness.

Physical and mental health status is assessed using the PROMIS®-29 survey instrument [25]. This measure assesses pain intensity using a single 0–10 numeric rating item, and seven health domains (physical function, fatigue, pain interference, depressive symptoms, anxiety, ability to participate in social roles and activities, and sleep disturbance) using five response options per domain ranging from "not at all" to "very much". In prior studies, this measure exhibits high reliability and validity, correlating well with other physical and mental health surveys as well as with chronic disease. The PROMIS® instrument assesses health-related quality of life over the past seven days, except for two domains (physical function and ability to participate in social roles and activities) which do not specify a timeframe. Raw PROMIS®-29 scores are re-scaled from raw scores of 8 (worst) to 40 (best) into a standardized T-score with a mean of 50 and a standard deviation (SD) of 10. A higher PROMIS® T-score represents more of the concept being measured. For negatively worded concepts like Anxiety, a higher score is worse. For positively worded concepts like Physical Function-Mobility, a higher score is better. Version 2.1 of the PROMIS®-29, used in the present study, is rescaled into a generic, societal, preference-based summary score [26]. This is based on PROMIS® scores for Cognitive Function, Abilities, Depression, Fatigue, Pain Interference, Physical Function, Sleep Disturbance, and Ability to Participate in Social Roles and Activities. It is scaled from 0 (equal to death) to 1 (equal to perfect health). Version 2.0 PROMIS® scores can also be used to estimate a Health Utility Index Mark 3 preference score [27]. In other settings, differences of 0.03 to 0.1 in this preference score have been interpreted as being clinically important [28–30]. A cutoff of 0.7 is used to determine severe impairment [31].

Cognition is assessed using the Patient-Reported Outcomes Measurement Information System (PROMIS®) Cognitive SF 8 [32]. PROMIS® Cognitive scores are re-scaled from raw scores of 8 (worst) to 40 (best) into standardized T-scores with a mean of 50 and standard deviation (SD) of 10. A higher score is interpreted as indicating greater cognition.

Health care process measures are assessed as ambulatory care and/or emergency department (ED) visits for symptoms related to COVID-19 illness as well as hospitalization (admitted to hospital overnight during study follow-up) as determined by data from the EMR.

Hospital-free and intensive care unit (ICU)-free survival are assessed as determined by data from the EMR [33]. Additionally, follow-up surveys include questions on COVID-19 related outpatient visits, emergency department visits, hospitalizations and ICU admissions. Hospital-free survival is defined as survival without any hospitalizations, and ICU-free survival as survival without time spent in the ICU.

Vaccination status, vaccination type, and timing of vaccination is assessed in the baseline and follow-up surveys. We will also use EMR data to validate vaccination status in a subset of enrolled participants.

Additional outcomes assessed include post-infectious sequelae (i.e., dyspnea, cough) [34–36], post-traumatic stress disorder (PC-PTSD-5) [37], and exercise (exercise vital sign; 2 question survey to assess habitual physical activity) [38–40], using previously validated questionnaires.

Social determinants of health (e.g., housing, available social services, geographical location, and education) are assessed using a previously validated questionnaire [41].

Work and activity status are assessed using questions about returning to work, missed work and activity level (S1 Appendix).

Myalgic encephalomyelitis/chronic fatigue syndrome (ME/CFS) is assessed using the CDC Short Symptom Screener (S1 Appendix), although there are no validated surveys to assess

ME-CFS. This survey, however, closely aligns with the 2015 Institute of Medicine diagnostic criteria for ME-CFS [42]. While the study design will allow for assessment of a range of clinical outcomes, ME-CFS was of particular interest given early reports that post-COVID sequelae may overlap with ME/CFS.

## Data sources

Results for a SARS-CoV-2 viral test (i.e., nucleic acid amplification test or antigen test) are confirmed by the research staff either through visualizing the result in the EMR or by reviewing an image of the test result sent to the research staff by the participant. After an eligible patient enrolls in the study, a combination of self-reported information and information generated from the patient's own health information is connected to the Hugo platform, collected, and sent to the study site over the 18-month follow-up period (S1 Appendix).

**Self-reported data.**   Using Hugo, surveys including the variables outlined above are sent by electronic mail or text to participants at the research site every 3 months throughout the follow-up period (Fig 1). These responses are sent through the Hugo platform and then shared with the study team. Subjects use their personal smartphone, tablet, computer, or other electronic device to connect to the internet and answer surveys that ascertain their symptoms, health care use, and physical, mental, and social health over an 18-month period. Responses sent by participants are encrypted. To minimize participant burden, each data collection episode is designed to take approximately 15 minutes or less to complete.

**Patient-centric data sharing.**   At enrollment, participants connect their health system portal account/s with the Hugo platform (Fig 2). Hugo is a web-based platform used to longitudinally collect data for this study. The Hugo platform gives patients the ability to collect and maintain their personal health records in a centralized, cloud-based account. Participants create a Hugo account and connect the health system portal accounts they choose to connect with the Hugo platform. This may include patient portals from healthcare systems, pharmacies, laboratories, and insurers. Individuals direct Hugo to share their health records with the research team according to the terms in the informed consent. This information is sent from Hugo to the research analytic core and stored in accordance with their institutional policies. The specific portal-related information available through Hugo in use for INSPIRE includes all data available through the patient portal, including medications, appointments and visits, test orders and results, clinical notes, problems, diagnoses, vital signs, demographics, and immunizations. After the study, participants can maintain their Hugo account or opt to delete their account and data.

Participants can connect their health system(s) portal accounts with Hugo at set up but may need assistance if they have technical issues or if they do not complete this initial step at enrollment. Once participants create an account in Hugo and link their portals, no additional actions are required to stream data into Hugo. Technical support from the enrolling site or clinical core is provided to resolve any difficulties setting up an account.

During the study, periodically Hugo downloads identifiable data outlined in the IRB protocol and consent form. Research sites have access to site-specific dashboards to track enrollment, identify which data sources are connected, and to monitor survey responses. Deidentified, individual-level data are sent from Hugo to the analytic team periodically for quality assurance and analysis.

## Human subjects considerations

This study involves self-enrollment with an online consent process using an electronic consent form designed with easy-to-read language. The consent form is a click-through digital

| | Enrolment | Post-enrollment | | | | | | Close-out |
|---|---|---|---|---|---|---|---|---|
| TIMEPOINT | 0 | 3 mos. | 6 mos. | 9 mos. | 12 mos. | 15 mos. | 18 mos. | Exit |
| ENROLLMENT: | | | | | | | | |
| Eligibility screen | X | | | | | | | |
| Informed consent | X | | | | | | | |
| ASSESSMENTS: | | | | | | | | |
| Age, gender, symptoms of SARS-CoV-2 infection, recent health care use, social determinants of health | X | | | | | | | |
| Comorbidities | | X | | | | | | |
| Outpatient visit, emergency department visit, hospitalization, intensive care stay, death, physical and mental health (PROMIS-29), cognitive SF-8, hospital-free survival, intensive care-free survival, dyspnea, cough, exercise, social determinants of health, work and activity status, myalgic encephalomyelitis/chronic fatigue syndrome | | X | X | X | X | X | X | X |

**Fig 1. Participant timeline.**

document where, after reading the document, the participant clicks to agree or disagree to study participation.

Electronic consent occurs through the Hugo platform. Participants are eligible to receive a small incentive for completion of each periodic questionnaire; the total value will be $100 over the course of the study. Researchers will not provide any information gathered through the study to clinicians engaged in treating the patient. Patients will be informed and reminded that their responses will not be provided to their healthcare team, both at the beginning of the study during the consent process and throughout the study on the regular questionnaires.

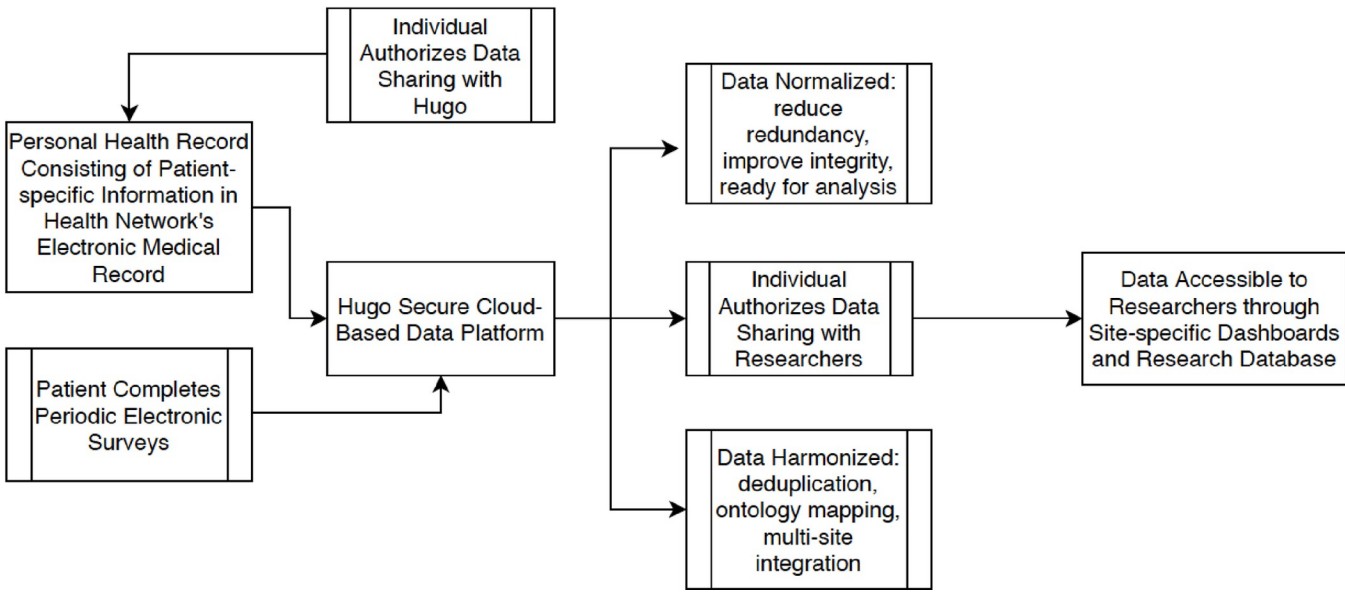

**Fig 2. Use of Hugo health to capture participant data.**

Ethics approval of this protocol has been obtained at each individual site including Rush University (protocol number: 20030902, approved 3/14/2020), Yale University (2000027976, approved 4/30/2020), the University of Washington (UW Human Subjects Division, STUDY00009920, approved 4/2/2020), Thomas Jefferson University (20p.1150, approved 1/21/2021), the University of Texas Southwestern Medical Center (STU 2020–1352, approved 2/3/2021), the University of Texas, Houston (HSC-MS-20-0981, approved 9/10/2020), the University of California, San Francisco (20–32222, approved 1/25/2021) and the University of California, Los Angeles (20–001683, approved 12/18/2020). The Yale University ethics approval includes the role as the analytic lead. Additionally, the Rush University ethics approval includes INSPIRE data storage on the Hugo platform and transfer of data to Rush for secure storage.

## Study size

Our target enrollment at study initiation was 3,600 people with SARS-CoV-2 infection confirmed by a positive SARS-CoV-2 viral test (i.e., nucleic acid amplification test or antigen test) and 1,200 people with a negative SARS-CoV-2 viral test. We expect that the age distribution of enrolled subjects will be broadly representative of patients tested in each site. Four of the sites (Yale, Jefferson, Rush and UW) draw on smaller catchment populations so have planned enrollment of 400 subjects per site. Four sites (UT Southwestern, UT Houston, University of California, San Francisco, and University of California, Los Angeles) have larger catchment populations so have planned enrollment of 800 subjects per site. Due to heterogeneity in baseline characteristics among study subjects identified early in the study, we modified the enrollment to support a 1:1 case/control ratio at each site, to better enable the comparison of 'like' cases with 'like' controls.

We estimated the power to detect relative differences in outcome rates between those who are symptomatic and test positive for SARS-CoV-2 as compared to those who are symptomatic but test negative for SARS-CoV-2. These power calculations are agnostic to the outcome of interest but based on the aim to examine relative differences in long-term outcomes between

**Table 4. Event rate difference between groups.**

| Power | 0.07 vs 0.05 | | | 0.10 vs 0.05 | | | 0.15 vs 0.05 | | | 0.20 vs. 0.05 | | |
|---|---|---|---|---|---|---|---|---|---|---|---|---|
| | Total N | COVID (+) | COVID (-) | Total N | COVID (+) | COVID (-) | Total N | COVID (+) | COVID (-) | Total N | COVID (+) | COVID (-) |
| 0.9 | 8012 | 6009 | 2003 | 1592 | 1194 | 398 | 516 | 387 | 129 | 180 | 135 | 45 |
| 0.8 | 6076 | 4557 | 1519 | 1224 | 918 | 306 | 404 | 303 | 101 | 144 | 108 | 36 |
| 0.7 | 4844 | 3633 | 1211 | 988 | 741 | 247 | 332 | 249 | 83 | 120 | 90 | 30 |
| 0.6 | 3904 | 2928 | 976 | 808 | 606 | 202 | 272 | 204 | 68 | 100 | 75 | 25 |

individuals with and without SARS-CoV-2 based on and between age strata. Prior to changing the enrollment strategy to a 1:1 case/control ratio, we reran the power analysis. These findings showed that with 1:1 enrollment, we had increased power to find a difference between SARS-CoV-2 infected vs. uninfected participants. For comparison of ME/CFS incidence in SARS-CoV-2 infected vs. uninfected, power calculations are based on the null hypothesis that there is no difference between individuals with and without SARS-CoV-2 in the outcome rate.

The assumptions used to generate these power calculations include both elements outside the study's team's control (e.g., baseline outcome rate in individuals without SARS-CoV-2) as well as elements amenable to changes in study design (e.g., individuals with SARS-CoV-2 strata group size). Assumptions include:

- 3,600 individuals with SARS-CoV-2 and 1,200 individuals without SARS-CoV-2

- Baseline outcome rate in individuals without SARS-CoV-2: 2.5% with the contingency that the baseline outcome rate may vary between age strata (18–40 years, 41–64 years, $\geq 65$ years)

- Outcome rate in individuals with SARS-CoV-2: presented as scenarios based on absolute or relative differences from baseline outcome rate in individuals without SARS-CoV-2. Also presented under the likely scenario that outcome rates vary between age strata

- Alpha = 0.05 (fixed)

Under conservative assumptions, the planned sample has 97.8% power to detect an absolute outcome rate difference of 2.5% between individuals with versus without SARS-CoV-2 infection. The power to detect a relative difference in outcome rates is highly sensitive to both the actual baseline outcome rate as well as variation in baseline outcome rates between age strata. The study would have adequate power (conventionally defined as 0.8) to detect a difference as small as 5% in outcomes with 1,224 total participants (25.5% of total planned enrollment) enrolled in a 3:1 ratio, or 918 COVID+ and 306 COVID negative (Table 4).

## Statistical methods

Statistical analyses will describe disease course and outcomes, including the characterization of specific symptoms and duration of symptoms (including duration of symptoms prior to enrollment), health care utilization (emergency department, hospitalization, post-acute care) with clinical (morbidity/mortality) and patient-reported health status outcomes as well as recovery (early and late disease sequelae). Baseline and demographic characteristics will be summarized by descriptive summaries (e.g., means and standard deviations for continuous variables such as age and percentages for categorical variables such as gender, medians, and quartiles for skewed data) and will be assessed by site of enrollment and compared to regional demographic data to evaluate for potential selection bias relative to site-specific patient populations and to assess for representativeness of the local population. Analyses will:

- Compare health status at baseline and follow-up between persons in the same age group who test SARS-CoV-2 positive and negative at initial test.

- Characterize health care utilization (ambulatory and ED visits, hospitalizations, post-acute care, telehealth visits) among SARS-CoV-2 positive participants by age and compare these to SARS-CoV-2 negative participants.

- Characterize and compare health outcomes by age and SARS-CoV-2 status: emergency or ambulatory care, admission to hospital); ICU-free survival; hospital-free survival; and subsequent patient-reported health status (cognition; physical health; mental health; and return to work).

We will use statistical modeling to estimate the association between key covariates and outcomes, including evaluating interactions by age. We plan to use survival analysis techniques to analyze time to outcome events, logistic regression for binary outcomes (e.g., hospitalization versus no hospitalization), Poisson or Cox regression for count data (e.g., number of hospitalizations over time), and linear regression models for continuous outcomes (e.g., PROMIS-29). Multiple imputation will be considered to handle missing covariate and outcome data in these analyses [43]. Chained equations will be used to impute each variable with missing data. Sensitivity analyses will be conducted using missing categories for covariates and including only people with non-missing outcome information.

We will use statistical analyses to compare the risk of ME/CFS and other health conditions in those with versus without SARS-CoV-2 infection as risk differences, risk ratios or odds ratios, as appropriate. Depending on the specific analysis question, we will also match or adjust for additional patient-level factors of interest, such as age, sex, race/ethnicity, income, and presence of specific underlying conditions such as hypertension and diabetes. Additional covariates of interest available from electronic record data will be assessed for their association with adverse outcomes.

## Data monitoring

Periodic reports of de-identified data from participants at each site are exported from the Hugo data management system for monitoring and analysis by the study cores. These data will be used to monitor the adequacy of recruitment, with due consideration to the balance of subjects with positive versus negative tests as well as their age distribution, and completeness of follow-up questionnaires.

## Reporting

The results of this study will be reported using the STrengthening the Reporting of OBservational studies in Epidemiology (STROBE) guidelines for reporting observational studies [44]. We intend to disseminate the results as rapidly as possible to help contribute to the COVID-19 response.

## Research partnership

This study is conducted in collaboration with colleagues from the CDC. Scientists from the CDC assisted with study design and will remain engaged as we evaluate and disseminate the research findings.

## Discussion

### Expected key results

Our prospective cohort study is designed to gain critical information needed by clinicians in the USA and globally regarding medium and long-term sequelae of SARS-CoV-2 infection.

Prior limited experience with people infected with SARS-CoV-2 suggests that age as well as race and ethnicity are important risk factors for SARS-CoV 2 infection and for poor outcomes in the acute setting [20, 45–48]), but there remains limited information concerning predictors of medium- and long-term outcomes. Moreover, there is a need to leverage digital assets and tools to harness data from both patients and providers to quickly define the clinical epidemiology of COVID-19. Creating such a platform and assessing its value to address priority questions would have immense significance for the nation.

To date, there has been a relative lack of epidemiological studies designed to provide robust evidence of incidence, risk factors, and natural history of sequelae of SARS-CoV-2 infection. Existing studies are limited by multiple potential biases in study design. Many prior studies of SARS-CoV-2 infection recruited from single rather than multiple centers, which may overestimate the relationship between baseline factors as well as treatment exposures and outcome and thereby lack generalizability [49–51]. In addition, many have preferentially recruited patients who are hospitalized or receive intensive care, who have greater acuity or severity of illness and could be at greater risk of long-term sequelae than those who are treated in an ambulatory setting. In contrast, our recruitment of participants testing for SARS-CoV-2 with acute symptoms of illness from community (e.g. drive up testing sites), clinic, emergency, and in-patient settings reduces the likelihood of selection bias [52].

Prior studies which lack concurrent controls who have similar symptoms or require similar health care utilization characterize the burden of SARS-CoV-2 infection, but, by the nature of their design, cannot describe the relative risk of sequelae compared to those who do not have SARS-CoV-2 infection but who have another viral infection. Social isolation and receipt of intensive care are both associated with initial and long -term social, emotional, well-being and health sequelae [53, 54]. The effects of increased social isolation affecting individuals' social, emotional, and functional well-being regardless of infection with SARS-CoV-2, further emphasizes the need for including individuals not infected with SARS-CoV-2 to generate evidence on the incidence of sequelae of COVID-19 illness.

Increasing evidence suggests that a significant proportion of individuals infected with SARS-CoV-2 experience ongoing clinical symptoms several months after acute infection. Among a cohort of 4,182 participants testing positive for SARS-CoV-2 evaluated in the COVID Symptom Study with the majority from the UK (88%), the US (7%), and Sweden (4.5%), 13% reported symptoms greater than 28 days after onset, with fatigue (98%) and headaches (91%) the most commonly reported symptoms [55]. Additionally, 4.5% of all participants had $\geq$ 56 days of symptoms and 2.6% had $\geq$ 84 days of symptoms [55]. In a large prospective cohort study from Wuhan, China, with in-person evaluations of 1,733 patients at 6 months from symptom onset, 76% of patients reported at least one symptom with the most commonly reported symptoms being fatigue/muscular weakness (63%), sleep disturbance (26%), and anxiety/depression (23%) [56]. Among a cohort of 180 participants in the Faroe Islands who had reverse transcription-polymerase chain reaction positive COVID-19 tests, 53% reported persistence of at least one symptom after a mean of 125 days from symptom onset [57]; the most persistent severe symptoms reported were fatigue, loss of smell and taste, and arthralgias [57]. In Italy, among 143 patients discharged from the hospital after acute COVID-19 with a mean follow-up of 60 days from first symptom onset, 87% of patients had persistent symptoms with the predominant symptoms being fatigue (53%), dyspnea (43%), arthralgia (27%), and chest pain, (22%) and with 44% experiencing a worsening quality of life [58]. A US-based study using historical comparator groups with viral lower respiratory tract illness and propensity matching demonstrated that 14% of adults $\leq$ 65 infected with SARS-CoV-2 had at least one new clinical sequelae requiring medical care after the acute phase of the illness; this was nearly 5 percentage points higher than the historical comparator group [59].

Similar findings of persistent symptoms have been reported in other studies across the world [12, 60].

Studies have also reported changes in renal function, metabolic response, cardiovascular systems, and neurological changes persisting in the post-acute phase of SARS-CoV-2 infection [61]. Of note, a considerable proportion of those previously infected with SARS-CoV-2 develop new clinical sequelae that were not present during the acute illness and that require medical attention; new clinical sequelae include chronic respiratory failure, cardiac arrhythmia, hypercoagulability, encephalopathy, peripheral neuropathy, amnesia, diabetes, liver test abnormalities, myocarditis, anxiety and fatigue [62]. There remains uncertainty regarding the incidence of sequelae, the range of impacts on individuals, the factors associated with risk for development of sequelae, and the natural history/time course of the sequelae. Given the high global rate of SARS-CoV-2 infection, this information is critical to guide care following SARS-CoV-2 infection amongst vast numbers of individuals. This evidence could potentially highlight individuals at higher risk for complications who may require more intensive follow-up and facilitate earlier intervention and guide targeting of primary or secondary interventions [63]. Further, dissemination of findings from this research may increase COVID-19 vaccination acceptance, helping to prevent sequelae from SARS-CoV-2 infection.

## Strengths and limitations

Our cohort study integrates detailed self-reports of participants with automated capture of their EHR information to provide more comprehensive data than traditional approaches. This innovative method enables capturing of baseline medical conditions, accounting for intermediate events between SARS-CoV-2 infection and sequelae, and objective assessments over time. Merging self-reported and digital data in this manner paves the way for similar research into long-term sequelae in other disease entities, including non-infectious illnesses such as trauma.

Using the on-line digital platform to collect patient-oriented outcomes enables us to adapt the survey content in real time as new information emerges. For example, we adapted the survey when vaccines became available. Additionally, there is the potential for this cohort to be engaged in future research as new questions arise relating to long-term sequalae, therapies and other related issues. Recruitment design strengths include the inclusion of participants with a range of disease severity, including participants with and without a history of hospitalization. Additionally, we seek to recruit participants who are ethnically diverse and geographically dispersed. Further, our design includes concurrent controls with negative SARS-CoV-2 diagnostic results to overcome the variable exposure to healthcare access as well as to COVID-19 mitigation strategies implemented in the community (e.g., masking mandates, shut-downs) which could otherwise bias assessment of the risk of patient-reported outcomes. Finally, our planned sample size will enable detection of rare events among the study population.

Our design anticipates and addresses study limitations as feasible. First, inclusion in the INSPIRE study requires participants to have a sufficient degree of technology literacy as well as periodic access to the internet which may introduce selection bias into who is able to participate. There are large differences in the use of desktop or laptop computers between Black (58%) or Hispanic or Latino persons (57%) vs. White (82%) individuals [64]. Importantly, about 80% of Black, Hispanic or Latino persons, and White adults own a smartphone; lack of availability of a smartphone should not be a barrier to enrollment of racial/ethnic subgroups in this study. We worked to overcome challenges related to technology literacy by offering support at the time of enrollment and throughout the study to troubleshoot problems such as linking digital health portals to the Hugo platform and completing quarterly surveys. Second,

while medium and long-term sequelae of SARS-CoV-2 infection can occur following asymptomatic infection, we made the decision not to enroll asymptomatic individuals and thus will not be able to report on outcomes in this group. We anticipate that individuals with COVID-19 and symptomatic illness are more likely to have the outcomes of interest. Third, there is the potential for selection bias in which participants whose COVID-19 illness has not yet resolved are more likely to enroll in this study. To mitigate against this, we required that enrollment occurwithin 42 days of SARS-CoV-2 infection diagnosis. At the analysis stage, sensitivity analysis can describe long-term symptoms as a function of time between onset and enrollment.

Fourth, given the prospective longitudinal design of this study, there is an inherent risk of attrition and loss-to-follow-up of participants. To limit this potential problem, each research site monitors research participants' progress and invites them to re-engage if they do not complete their quarterly survey. While the preferred means of contact varies between sites, this may include through email, short message service (i.e., text), or telephone reminder. Additionally, participants are incentivized to participate with a monetary reward for survey completion as described above (consent and ethics section).

Fifth, there is a risk of poor data quality and misclassification. With this study design which incorporates digital health data, some data is passively acquired and thereby data quality is dependent on the linkage to and capture of electronic medical records. We believe that coupling these digital health data with the self-reported data will enhance the accuracy and completeness of available information. There is also potential for bias from unmasking underlying health conditions identified during the study that were not known until after the SARS-CoV-2 infection resulting in misclassification of pre-existing diagnoses as sequelae of COVID-19. As feasible, we will evaluate indications of undiagnosed health conditions by evaluating data from the EHR.

Sixth, is the risk of misclassification bias from inaccurate diagnostic test results of COVID-19 tests at enrollment. Despite the reported high sensitivity and specificity of SARS-CoV-2 diagnostic testing [65, 66], some participants could be misclassified as having or not having acute SARS-CoV-2 infection. Participants with false negative test results are classified as controls but can still have increased risks of long-term sequelae, which may bias our results towards the null. As well, participants' SARS-CoV-2 status can change over time due to repeat exposure and repeat diagnostic testing. We inquire about repeat SARS-CoV-2 testing and results during quarterly surveys and we will also look for related data during electronic medical record review.

## Conclusions

Upon the conclusion of the study, we will be able to quantify the burden of long-term SARS-CoV-2 sequelae as well as characterize predictors of sequelae. Additionally, data from INSPIRE will offer insight into syndromes with overlapping signs and symptoms, such as ME/CFS, as well as how people with underlying disease and subsequent COVID-19 experience these illnesses. We will be better poised to develop prevention and treatment strategies and to tailor these strategies for the most at-risk subsets of the population. The results will inform clinicians and public health authorities and will help prepare for future SARS-CoV-2 surges.

## Supporting information

**S1 File.**
(DOCX)

**S1 Appendix. INSPIRE investigators.**
(DOCX)

## Author Contributions

**Conceptualization:** Kelli N. O'Laughlin, Matthew Thompson, Bala Hota, Michael Gottlieb, Erica S. Spatz, Kari A. Stephens, Arjun Venkatesh, Sharon Saydah, Harlan M. Krumholz, Joann G. Elmore, Robert A. Weinstein, Graham Nichol.

**Data curation:** Harlan M. Krumholz.

**Funding acquisition:** Bala Hota, Robert A. Weinstein.

**Investigation:** Kelli N. O'Laughlin, Bala Hota, Michael Gottlieb, Anna Marie Chang, Lauren E. Wisk, Ralph C. Wang, Erica S. Spatz, Kari A. Stephens, Ryan M. Huebinger, Samuel A. McDonald, Arjun Venkatesh, Nikki Gentile, Benjamin H. Slovis, Mandy Hill, Ahamed H. Idris, Robert Rodriguez, Harlan M. Krumholz, Joann G. Elmore, Robert A. Weinstein, Graham Nichol.

**Methodology:** Kelli N. O'Laughlin, Matthew Thompson, Bala Hota, Erica S. Spatz, Kari A. Stephens, Arjun Venkatesh, Sharon Saydah, Harlan M. Krumholz, Joann G. Elmore, Robert A. Weinstein, Graham Nichol.

**Project administration:** Bala Hota, Robert A. Weinstein, Graham Nichol.

**Software:** Harlan M. Krumholz.

**Supervision:** Bala Hota, Robert A. Weinstein, Graham Nichol.

**Writing – original draft:** Kelli N. O'Laughlin, Matthew Thompson, Michael Gottlieb, Graham Nichol.

**Writing – review & editing:** Kelli N. O'Laughlin, Bala Hota, Michael Gottlieb, Ian D. Plumb, Anna Marie Chang, Lauren E. Wisk, Aron J. Hall, Ralph C. Wang, Erica S. Spatz, Kari A. Stephens, Ryan M. Huebinger, Samuel A. McDonald, Arjun Venkatesh, Nikki Gentile, Benjamin H. Slovis, Mandy Hill, Sharon Saydah, Ahamed H. Idris, Robert Rodriguez, Harlan M. Krumholz, Joann G. Elmore, Robert A. Weinstein, Graham Nichol.

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
