## [Decision Letter · Decision Letter 0]

2 Dec 2021

PONE-D-21-19569Study protocol for the Innovative Support for Patients with SARS-COV-2 Infections Registry (INSPIRE): a longitudinal study of the medium and long-term sequelae of SARS-CoV-2 infectionPLOS ONE

Dear Dr. O'Laughlin,

Thank you for submitting your manuscript to PLOS ONE. After careful consideration, we feel that it has merit but does not fully meet PLOS ONE’s publication criteria as it currently stands. Therefore, we invite you to submit a revised version of the manuscript that addresses the points raised during the review process.

We look forward to receiving your revised manuscript.

Kind regards,

Gerald Chi, M.D.

Academic Editor

PLOS ONE

Journal Requirements:

2. Thank you for stating the following in the Competing Interests/Financial Disclosure section:

“I read the journal's policy and the authors of this manuscript have the following competing interests: HMK is co-founder for HugoHealth.”

We note that one or more of the authors are employed by a commercial company: HugoHealth.”

3. We note you have included a table to which you do not refer in the text of your manuscript. Please ensure that you refer to Table 3 in your text; if accepted, production will need this reference to link the reader to the Table.

Reviewers' comments:

Reviewer's Responses to Questions

**Comments to the Author**

1. Does the manuscript provide a valid rationale for the proposed study, with clearly identified and justified research questions?

Reviewer #1: Yes

Reviewer #2: Yes

2. Is the protocol technically sound and planned in a manner that will lead to a meaningful outcome and allow testing the stated hypotheses?

Reviewer #1: Yes

Reviewer #2: Yes

3. Is the methodology feasible and described in sufficient detail to allow the work to be replicable?

Reviewer #1: Yes

Reviewer #2: No

4. Have the authors described where all data underlying the findings will be made available when the study is complete?

Reviewer #1: Yes

Reviewer #2: Yes

5. Is the manuscript presented in an intelligible fashion and written in standard English?

Reviewer #1: Yes

Reviewer #2: Yes

6. Review Comments to the Author

You may also provide optional suggestions and comments to authors that they might find helpful in planning their study.

Reviewer #1: O'Laughlin and colleagues have presented a robust, well-defined study protocol to examine long-term sequelae of COVID-19 infection. They acknowledge the limitations of their proposed data collection methods (notably the possibility of under-representation of racial and ethnic minorities due to technological accessibility) but presently there are no feasible alternatives on the scale necessary for this study.

Reviewer #2: The study addresses a very important issue in trying to provide standardize and comparative information regarding medium and long-term sequelae of SARS-CoV-2 infection.

Given the nature of patient enrollment, the study has the inherent risk of potential biases whose planned management have been only partially explained in the text.

Remaining questions are:

-In page 16 it is said that it is expected that "the distribution of enrolled subjects will represent that of patients tested in each site" but is this going to be specifically analyzed? (e.g comparing the population characteristics between tested individuals at sites with patients enrolled in the study)

-Given the possibility for selection bias according to age, is any age range cap planned?

-Patients enrolled through phone, mail or personal invitation may differ, is any plan to stratify enrollment according to entry procedure? Controls will be matched by site?

-It is very likely that a significant proportion of the enrolled population may have already received SARS-COV-2 specific vaccination prior to study entry. This may affect both the number and magnitude of the symptoms. Is there any plan to adjust the analysis according to vaccination status?

-Symptoms may also vary according to the prevalent variant of SARS-CoV-2 virus. Is there any plan to assess specific strains and adjust the results accordingly?

-Is there any plan to adjust for duration of symptoms prior to enrollment within the allowed 42 days after diagnosis? Some patients may have persistant symproms whereas other may have been asymptomatic for days or weeks until they develop new symptoms

-It is not entirely clear whether 1 or 2 symptoms listed in Table 2 will be required for study entry.

7. PLOS authors have the option to publish the peer review history of their article (what does this mean?). If published, this will include your full peer review and any attached files.

Reviewer #1: **Yes: **Catherine Chen

Reviewer #2: No

---

## [Author Response · Author response to Decision Letter 0]

30 Dec 2021

1. Comments from Reviewer: In page 16 it is said that it is expected that "the distribution of enrolled subjects will represent that of patients tested in each site" but is this going to be specifically analyzed? (e.g., comparing the population characteristics between tested individuals at sites with patients enrolled in the study)

We will compare the demographic characteristics of enrolled patients to those of the region of the site in which they were enrolled. We modified the manuscript on page 20 to read:

Baseline and demographic characteristics will be summarized by descriptive summaries (e.g., means and standard deviations for continuous variables such as age and percentages for categorical variables such as gender, medians, and quartiles for skewed data) and will be assessed by site of enrollment and compared to regional demographic data to evaluate for potential selection bias relative to site-specific patient populations and to assess for representativeness of the local population.

2. Comments from Reviewer: Given the possibility for selection bias according to age, is any age range cap planned?

We will enroll adults of any age and are not restricting enrollment to adults aged less than any particular cutoff. This is demonstrated in the inclusion criteria in Table 1. Selected analyses may restrict to a particular cutoff based on the objective. We believe this approach will allow for sufficient enrollment across all ages. At the point of our initial data freeze (enrollment of N=998), the mean age was 41 (standard deviation +/- 15.2); the minimum age was 18 and the maximum age was 88. We modified the text on page 18 to read:

We expect that the age distribution of enrolled subjects will be broadly representative of participants tested in each site.

Additionally, we modified the statistical methods section on page 21 to read:

We will use statistical analyses to compare the risk of ME/CFS and other health conditions in those with versus without SARS-CoV-2 infection as risk differences, risk ratios or odds ratios, as appropriate. Depending on the specific analysis question, we will also match or adjust for additional patient-level factors of interest, such as age, sex, race/ethnicity, income, and presence of specific underlying conditions such as hypertension and diabetes. Additional covariates of interest available from electronic record data will be assessed for their association with adverse outcomes. 

3. Comments from Reviewer: Patients enrolled through phone, mail or personal invitation may differ, is any plan to stratify enrollment according to entry procedure? Controls will be matched by site?

We wish to clarify that we recruit participants through phone, mail, personal invitation or social media. However, all patients must enroll through an internet connection (e.g., smartphone, computer). We do not plan to stratify enrollment according to smartphone vs. computer. Ordinarily analyses will group or stratify by site, although the precise method of analysis will depend on the objectives. We modified the manuscript to make the recruitment and enrollment processes clearer on page 10:

Methods used to recruit potentially eligible participants vary by site, although each site applies the same eligibility criteria described in Table 1. Most sites screen for eligible participants among those tested for SARS-CoV-2 infection. We seek to enroll participants as close to their initial date of SARS-CoV-2 testing as possible in order to reduce recall bias. Participant identification and recruitment methods include: i) participants learn of the study from a poster, brochure, or social media advertisement and/or ii) research staff identify potentially eligible individuals and reach out to them in-person, over the phone (e.g., text or call), or by e-mail to invite them to enroll. In some instances, members of the study team access the EMR to screen for eligible individuals based on SARS-CoV-2 testing and reason for testing. In other cases, contact information are obtained from organizations conducting SARS-CoV-2 testing. The recruitment methods used at each site are based on local IRB approval and practical considerations. Regardless of how the individual is recruited, all participants must enroll through the online portal. Individuals may access this portal through any device that connects to the internet (e.g., smart phone, tablet, computer).

4. Comments from Reviewer: It is very likely that a significant proportion of the enrolled population may have already received SARS-COV-2 specific vaccination prior to study entry. This may affect both the number and magnitude of the symptoms. Is there any plan to adjust the analysis according to vaccination status?

When we began enrollment in this study, SARS-COV-2 vaccinations were not available. As they became available, we expanded our follow-up questionnaires to elicit information about vaccination status. Please note that the Appendix A survey specifications document includes a detailed version history with dates and details of all changes. The COVID-19 vaccination questions were added on March 11, 2021. We modified the manuscript on page 13 in response to this comment. The text reads: 

Vaccination status, vaccination type, and timing of vaccination is assessed in the baseline and follow-up surveys. We will also use EMR data to validate vaccination status in a subset of enrolled participants.

5. Comments from Reviewer: Symptoms may also vary according to the prevalent variant of SARS-CoV-2 virus. Is there any plan to assess specific strains and adjust the results accordingly?

We agree that this may be the case. Although we are not collecting sequencing or other variant-screening information as part of the study, we will be able to assess differences in symptoms by calendar time. This will allow comparison of periods during which particular variants were dominant. The shift to Delta strains occurred over such a short period of time (that can be ascertained reliably by locale) that we may have an opportunity to impute likely strain type by time of disease onset. This may also be the case for Omicron strains. 

6. Comments from Reviewer: Is there any plan to adjust for duration of symptoms prior to enrollment within the allowed 42 days after diagnosis? Some patients may have persistent symptoms whereas other may have been asymptomatic for days or weeks until they develop new symptoms.

Depending on the objective of the analysis, we may adjust for duration of symptoms. We amended the manuscript “Statistical methods” section on page 20 accordingly to read:

Statistical analyses will describe disease course and outcomes, including the characterization of specific symptoms and duration of symptoms (including duration of symptoms prior to enrollment), health care utilization (emergency department, hospitalization, post-acute care) with clinical (morbidity/mortality) and patient-reported health status outcomes as well as recovery (early and late disease sequelae). Baseline and demographic characteristics will be summarized by descriptive summaries (e.g., means and standard deviations for continuous variables such as age and percentages for categorical variables such as gender, medians, and quartiles for skewed data).

Additionally, as noted in #2 above, we modified the statistical methods section on page 21 to read:

We will use statistical analyses to compare the risk of ME/CFS and other health conditions in those with versus without SARS-CoV-2 infection as risk differences, risk ratios or odds ratios, as appropriate. Depending on the specific analysis question, we will also match or adjust for additional patient-level factors of interest, such as age, sex, race/ethnicity, income, and presence of specific underlying conditions such as hypertension and diabetes. Additional covariates of interest available from electronic record data will be assessed for their association with adverse outcomes.

7. Comments from Reviewer: It is not entirely clear whether 1 or 2 symptoms listed in Table 2 will be required for study entry.

We wish to clarify that at least one symptom is required in order to be eligible for enrollment.

We modified Table 1 on page 8 in response to this comment to make this clearer. The related inclusion criteria now reads:

c) At least one self-reported symptom(s) suggestive of acute SARS-CoV-2 infection [19]

In addition, we changed Table 2 to be a complete list of the symptoms used to assess eligibility. We inserted this new Table 2 on page 9.

 

In addition to the specific changes made in response to the Reviewer comments, to make the manuscript succinct, clear and to ensure it reflects the most recent information in this changing landscape, we made the following changes:

8. We inserted Table 3 on page 11 to clearly demonstrate the data variables collected, the instrument sources and the schedule of survey delivery. While this information was inserted initially in the Appendix, we think improving this Table 3 enhances the manuscript by making the INSPIRE study methods clearer. 

9. Abstract: we made minor edits to the Methods section of the Abstract on page 3 to make the language more concise. We made the following changes:

INSPIRE is a prospective, multicenter, longitudinal study of individuals with symptoms of SARS-CoV-2 infection in eight regions across the US. Adults are eligible for enrollment if they are fluent in English or Spanish, reported symptoms suggestive of acute SARS-CoV-2 infection, and if they are within 42 days of having a SARS-CoV-2 viral test (i.e., nucleic acid amplification test or antigen test), regardless of test results. Recruitment occurs in-person, by phone or email, and through online advertisement. A secure online platform is used to facilitate the collation of consent-related materials, digital health records, and responses to self-administered surveys. Participants are followed for up to 18 months,…

10. We updated the Introduction on page 5 to reference more recent COVID-19 related data:

As of December 2021, > 51 million COVID-19 cases and > 805,000 attributed deaths have been detected in the USA [1]. Globally, > 276 million COVID-19 cases and > 5.3 million attributed deaths have been reported [2].

11. We revised the first introduction paragraph on page 5 on post-acute COVID-19 definitions to reflect recent updates: 

According to the Centers for Disease Control, post-acute COVID-19 is defined as emergent, recurring or persistent symptoms occurring ≥ 4 weeks following acute infection with COVID-19 (https://www.cdc.gov/coronavirus/2019-ncov/long-term-effects/index.html). Other sources describe post-acute COVID-19 as persistence of symptoms or development of sequelae after 3 or 4 weeks from the onset of acute symptoms of COVID-19 [8-10]. Some have divided the post-acute time period into subacute period (4-12 weeks) beyond acute COVID-19, and a chronic or post-COVID-19 syndrome (> 12 weeks), which includes symptoms persisting or present not attributable to alternative diagnoses [8,11]. Information on post-acute COVID-19 and long-term sequelae of SARS-CoV-2 infection has continued to emerge [12–15].

12. We inserted our revised plan to change from a 3:1 case/control enrollment target to a 1:1 case/control ratio and we explained the reason for this change. 

Related revisions were made in the Abstract (the Methods section) on page 3:

Our planned enrollment is 4,800 participants, including 2,400 SARS-CoV-2 positive and 2,400 SARS-CoV-2 negative participants (as a concurrent comparison group).

Related revisions were made in the Methods on the bottom of page 8:

We initially used a 3:1 case/control enrollment ratio to oversample those who are positive for SARS-CoV-2 on testing, while still ensuring an adequate control cohort for comparison. However, given the heterogeneity in baseline characteristics among study subjects, we modified the enrollment to support a 1:1 case/control ratio, which will better enable the comparison of ‘like’ cases with ‘like’ controls.

 Additionally, we revised the Methods section on page 18 to read:

Our target enrollment at study initiation was 3,600 people with SARS-CoV-2 infection confirmed by a positive SARS-CoV-2 viral test (i.e., nucleic acid amplification test or antigen test) and 1,200 people with a negative SARS-CoV-2 viral test. We expect that the age distribution of enrolled subjects will be broadly representative of patients tested in each site. Four of the sites (Yale, Jefferson, Rush and UW) draw on smaller catchment populations so have planned enrollment of 400 subjects per site. Four sites (UT Southwestern, UT Houston, University of California, San Francisco, and University of California, Los Angeles) have larger catchment populations so have planned enrollment of 800 subjects per site. Due to heterogeneity in baseline characteristics among study subjects identified early in the study, we modified the enrollment to support a 1:1 case/control ratio at each site, to better enable the comparison of ‘like’ cases with ‘like’ controls.

We estimated the power to detect relative differences in outcome rates between those who are symptomatic and test positive for SARS-CoV-2 as compared to those who are symptomatic but test negative for SARS-CoV-2. These power calculations are agnostic to the outcome of interest but based on the aim to examine relative differences in long-term outcomes between individuals with and without SARS-CoV-2 based on and between age strata. Prior to changing the enrollment strategy to a 1:1 case/control ratio, we reran the power analysis. These findings showed that with 1:1 enrollment, we had increased power to find a difference between SARS-CoV-2 infected vs. uninfected participants. For comparison of ME/CFS incidence in SARS-CoV-2 infected vs. uninfected, power calculations are based on the null hypothesis that there is no difference between individuals with and without SARS-CoV-2 in the outcome rate.

13. We edited the section on Myalgic encephalomyelitis/chronic fatigue syndrome (ME/CFS) to better explain the background regarding this aspect of the INSPIRE study. Further, we believe Table 3 is overly detailed and we recommend deleting it from the manuscript. We made these changes on pages 14-15:

Myalgic encephalomyelitis/chronic fatigue syndrome (ME/CFS) is assessed using the CDC Short Symptom Screener (Appendix A), although there are no validated surveys to assess ME-CFS. This survey, however, closely aligns with the 2015 Institute of Medicine diagnostic criteria for ME-CFS [42]. While the study design will allow for assessment of a range of clinical outcomes, ME-CFS was of particular interest given early reports that post-COVID sequelae may overlap with ME/CFS. 

14. We worked to make the limitations section clearer on pages 26-28 by inserting structuring words (i.e., First, Second, Third, etc.) and by revising the following section on page 27:

Second, while medium and long-term sequelae of SARS-CoV-2 infection can occur following asymptomatic infection, we made the decision not to enroll asymptomatic individuals and thus will not be able to report on outcomes in this group. We anticipate that individuals with COVID-19 and symptomatic illness are more likely to have the outcomes of interest. Third, there is the potential for selection bias in which participants whose COVID-19 illness has not yet resolved are more likely to enroll in this study. To mitigate against this, we required that enrollment occur within 42 days of SARS-CoV-2 infection diagnosis. At the analysis stage, sensitivity analysis can describe long-term symptoms as a function of time between onset and enrollment.

15. Deleted two sentences on page 22 referring to the Patient Advisory Board. While we intended at the study outset to create a Patient Advisory Board we have not yet conducted this activity to date. We prefer the protocol manuscript to reflect the activities to date and therefore we think it is best to delete these sentences.

16. Pg. 13, Methods, inserted more information about hospital-free and ICU-free survival:

Hospital-free and intensive care unit (ICU)-free survival are assessed as determined by data from the EMR [33]. Additionally, follow-up surveys include questions on COVID-19 related outpatient visits, emergency department visits, hospitalizations and ICU admissions. Hospital-free survival is defined as survival without any hospitalizations, and ICU-free survival as survival without time spent in the ICU.

17. Reviewed and updated citations. We reviewed our citations and updated the date accessed for each on-line citation accordingly.

18. Additionally, we made the following minor edits to make the language clearer:

Pg. 5, Introduction: To characterize post-COVID-19 syndromes better, there is an urgent need greater diversity in the study population to allow for representativeness and generalizability…

Pg. 12, Methods, inserted: Generic physical and mental health is assessed using the PROMIS®-29 survey instrument [25].

Pg. 12, Methods, inserted: Raw PROMIS®-29 scores are re-scaled from raw scores of 8 (worst) to 40 (best) into a standardized T-score with a mean of 50 and a standard deviation (SD) of 10.

Pg. 15, Methods, deleted: health care experience

Pg. 23, Discussion: Prior limited experience with people infected with SARS-CoV-2 suggests that age as well as race and ethnicity are important risk factors for SARS-CoV 2 infection and for poor outcomes in the acute setting [20,45–48]), but there remains limited information concerning predictors of medium- and long-term outcomes.

Pg. 22, Discussion: Increasing evidence suggests that a significant proportion of individuals infected with SARS-CoV-2 experience ongoing clinical symptoms several months after acute infection.

Pg. 26, Discussion: Finally, our planned sample size will enable detection of rare events among the study population.

---

## [Decision Letter · Decision Letter 1]

8 Feb 2022

Study protocol for the Innovative Support for Patients with SARS-COV-2 Infections Registry (INSPIRE): a longitudinal study of the medium and long-term sequelae of SARS-CoV-2 infection

PONE-D-21-19569R1

Dear Dr. O'Laughlin,

We’re pleased to inform you that your manuscript has been judged scientifically suitable for publication and will be formally accepted for publication once it meets all outstanding technical requirements.

Kind regards,

Gerald Chi, M.D.

Academic Editor

PLOS ONE

Reviewers' comments:

Reviewer's Responses to Questions

**Comments to the Author**

1. Does the manuscript provide a valid rationale for the proposed study, with clearly identified and justified research questions?

Reviewer #1: Yes

Reviewer #2: Yes

2. Is the protocol technically sound and planned in a manner that will lead to a meaningful outcome and allow testing the stated hypotheses?

Reviewer #1: Yes

Reviewer #2: Yes

3. Is the methodology feasible and described in sufficient detail to allow the work to be replicable?

Reviewer #1: Yes

Reviewer #2: Yes

4. Have the authors described where all data underlying the findings will be made available when the study is complete?

Reviewer #1: Yes

Reviewer #2: Yes

5. Is the manuscript presented in an intelligible fashion and written in standard English?

Reviewer #1: Yes

Reviewer #2: Yes

6. Review Comments to the Author

You may also provide optional suggestions and comments to authors that they might find helpful in planning their study.

Reviewer #1: The authors have satisfactorily responded to the critiques from the initial submission and made appropriate modifications in their manuscript.

Reviewer #2: Thank you for reviewing and sending a corrected version of the manuscript.

I believe all issues have been adequately addressed.

Would strongly suggest to consider a post-hoc study subanalysis according to vaccination status

7. PLOS authors have the option to publish the peer review history of their article (what does this mean?). If published, this will include your full peer review and any attached files.

Reviewer #1: No

Reviewer #2: No

---

## [Editor Report · Acceptance letter]

10 Feb 2022

PONE-D-21-19569R1 

Study protocol for the Innovative Support for Patients with SARS-COV-2 Infections Registry (INSPIRE): a longitudinal study of the medium and long-term sequelae of SARS-CoV-2 infection 

Dear Dr. O'Laughlin:

I'm pleased to inform you that your manuscript has been deemed suitable for publication in PLOS ONE. Congratulations! Your manuscript is now with our production department. 

Kind regards, 

on behalf of

Dr. Gerald Chi 

Academic Editor

PLOS ONE